# Nanomaterial Applications in Photothermal Therapy for Cancer

**DOI:** 10.3390/ma12050779

**Published:** 2019-03-07

**Authors:** Austin C.V. Doughty, Ashley R. Hoover, Elivia Layton, Cynthia K. Murray, Eric W. Howard, Wei R. Chen

**Affiliations:** 1Biophotonics Research Laboratory, Center for Interdisciplinary Biomedical Education and Research, College of Mathematics and Science, University of Central Oklahoma, Edmond, OH 73034, USA; adoughty1@uco.edu (A.C.V.D.); ahoover7@uco.edu (A.R.H.); elayton3@uco.edu (E.L.); 2Department of Mathematics and Statistics, College of Mathematics and Science, University of Central Oklahoma, Edmond, OH 73034, USA; cmurray@uco.edu; 3Department of Cell Biology, University of Oklahoma Health Sciences Center, Oklahoma City, OH 73104, USA; Eric-Howard@ouhsc.edu

**Keywords:** cancer, metastatic, photothermal therapy, ablation, nanomaterials, combination therapy

## Abstract

As a result of their unique compositions and properties, nanomaterials have recently seen a tremendous increase in use for novel cancer therapies. By taking advantage of the optical absorption of near-infrared light, researchers have utilized nanostructures such as carbon nanotubes, gold nanorods, and graphene oxide sheets to enhance photothermal therapies and target the effect on the tumor tissue. However, new uses for nanomaterials in targeted cancer therapy are coming to light, and the efficacy of photothermal therapy has increased dramatically. In this work, we review some of the current applications of nanomaterials to enhance photothermal therapy, specifically as photothermal absorbers, drug delivery vehicles, photoimmunological agents, and theranostic tools.

## 1. Introduction

The development of nanomaterials marks a significant step forward in photothermal therapy of cancers. Robust nanomaterials can now be designed to have specific optical, physicochemical, biological, and pharmaceutical properties to both compensate for the weaknesses and enhance the strengths of photothermal cancer therapy.

Cancer arises in the human body through the accumulation of genetic mutations in cellular DNA [1,2]. The mechanisms that regulate cell death and cell division become damaged, leading to uncontrolled multiplication of poorly functioning cells in the body. While this can take many forms, the establishment of cancerous cells quite commonly leads to the formation of tumor masses in the body [3]. When these masses remain small or are detected early, the most common and effective medical procedure is to simply resect the tumor from the body. However, tumors in some organs such as the brain or pancreas are quite difficult to remove without significant damage to healthy tissue nearby [4,5]. In addition, once the tumor masses have grown, the cancer cells often escape the original site in search of more nutrient-rich environments, forming distant metastases.

When the cancer is in an advanced stage or the tumors are deemed inoperable, other treatment strategies must be utilized. Radiotherapy and chemotherapy are the conventional oncologic methodologies; however, these are accompanied by an extreme reduction in patient quality of life [6,7]. As a result, much of the recent research in the field has been devoted to developing new treatment modalities with reduced side-effects for these difficult-to-treat cancers. Foremost among these are hormonal therapies [8] and various kinds of targeted therapies such as checkpoint-inhibitor therapy [9], photothermal therapy (PTT) [10,11], and photodynamic therapy (PDT) [12]. Hormonal therapies are designed to inhibit hormone-sensitive cancers of the endocrine system, principally of the breast, prostate, adrenal gland, or endometrium. Light-based therapies, both photothermal and photodynamic, are designed to selectively kill the cancerous tissue in the body through either thermal or oxidative stress, respectively.

PTT, in particular, shows strong promise for treating tumors. In PTT, usually a near-infrared (NIR) laser is used to illuminate the target tumor either topically or interstitially through an optical fiber, and the light energy is converted into heat through optical absorption. Over time, this process leads to either partial or complete ablation of the target tissue, depending on the PTT regime. Through the use of selective photothermal absorbers, difficult-to-treat tumors can be targeted with minimal invasiveness. Similarly, advanced cancers can be treated by utilizing partially-ablated tumors as a source of both immunological stimulation and tumor antigens.

The photothermal absorption in tumors is highly dependent on the photothermal transducer, the wavelength of light coming from the laser, and the mode of laser light delivery (either interstitial or non-invasive). All modes of laser light delivery in PTT aim to increase the temperature in a uniform manner in tumor tissues while preventing damage to healthy surrounding tissues. Photothermal damage of tumor cells typically commences when tumor temperature reaches 41 °C [13]. However, as effective ablation of the tumors requires the destruction of every cancer cell, PTT often requires the tumor center to reach higher temperatures (≥ 50 °C), and a temperature gradient will form such that the edge of the tumor will reach therapeutic temperatures [14,15,16]. In photoimmunotherapeutic applications, this temperature gradient provides an advantage as it provides a broader range of cancer expression within the tumor microenvironment [17].

The study of nanomaterials, materials with one dimension between 1 and 100 nm, is a burgeoning field of research, and applications range from industrial sensors to medical devices. These nanomaterials can have a variety of unique and specific properties that depend on their chemical structure, method of synthesis, and modification. The optical absorption spectra and biocompatibility of nanomaterials are of particular importance in photothermal medical applications. Many nanomaterials exhibit strong absorption in the NIR range and can thus act as effective photothermal transducers. For some nanostructures, altering the synthesis allows an absorption peak to be fine-tuned to a very narrow range of wavelengths. This increases the specificity of the photothermal effect and improves the quality of photothermal treatments. Similarly, many nanostructures can be conjugated to a variety of surface-modifying molecules such as polymers or antibodies. This can alter the nanomaterial’s biopersistence as well as ameliorate toxicological concerns regarding some nanostructures.

One of the primary challenges with PTT is that heat will inevitably seep out of the target tissue and damage the surrounding tissue. To curb this, photothermal absorbers are used to enhance the heat generation in the target tissue. With a photothermal absorber, less total light energy is required to achieve therapeutic temperatures, ultimately leading to less heat escaping from the target tumor and reduced damaged to the healthy surrounding tissue. The choice of photothermal absorber is paramount to maximize the treatment efficiency. Various small molecule optical dyes, such as indocyanine green, have been useful to this end [18]. In addition, many nanomaterials also serve as suitable photothermal absorbers. Nanomaterials also have the unique advantage of offering a dynamic platform for the design of an effective combination therapy.

In this paper, we review the overall application of nanomaterials to enhance PTT for cancer treatment. In Section 2, we discuss the use of nanomaterials as selective photothermal absorbers to target the photothermal effect to the tumor tissue and reduce the damage to healthy, surrounding tissues. This review covers the three most common nanomaterials: graphene oxide sheets, carbon nanotubes, gold nanomaterials, as well as overviewing some alternative nanostructures. In Section 3, we briefly discuss the use of nanomaterials as drug delivery vehicles, with a particular emphasis on photo-chemotherapy and photo-radiotherapy. We discuss the use of immunological agents with nanomaterials to instigate a systemic antitumor response in Section 4. In Section 5, we review theranostic applications of nanomaterials for combined diagnosis and treatment.

## 2. Nanomaterials as Selective Photothermal Absorbers

The fundamental application of nanomaterials to PTT is to enhance the photothermal selectivity of light absorption in the target tissue. Many different nanomaterials are suitable for this purpose, with structural chemistries ranging from simple constructions like colloidal gold to complex organic polymers.

Colloidal gold nanoparticles (GNPs), as either gold nanorods (GNRs), gold nanoshells [19], or gold nanocages [20], absorb NIR light through surface plasmon resonance [21]. By altering the synthesis conditions, the size and shape distribution of the nanoparticles can be controlled, and the resonant frequency adjusted. This leads to precise control over the absorption spectra of GNPs, allowing the nanoparticles to be fine-tuned to absorb light of a particular wavelength. As a result, PTT using gold nanomaterials exhibit excellent specificity.

As with all nanoparticles, the toxicological concerns of gold nanomaterials are of crucial importance prior to in vivo application. The synthesis of gold nanoparticles often involves the use of toxic surfactants like cetrimonium bromide (CTAB), and careful steps must be taken to ensure that no excess harm is introduced with the nanoparticles [22]. Most commonly, a thiolated polymer such as polyethylene glycol (PEG) is used to replace the CTAB on the GNP surface. The colloidal gold, itself, is highly biocompatible as it is inert in biological tissue [23].

Due to their exemplary optical and biological properties, GNPs have frequently been applied in PTT. Initially reported by El-Sayad et al. [24], GNRs have been used to selectively kill cancerous cells. In their initial report, El-Sayad conjugated anti-epidermal growth factor receptor (anti-EGFR) antibodies to the GNR surface. As EGFR is significantly overexpressed on the surface of malignant cancer cells, these nanoparticles could selectively localize within the tumor region, allowing subsequent ablation with a NIR laser. This was later demonstrated in vivo through both intratumoral and intravenous injection in a mouse with a xenograft head and neck cancer model [25]. Later groups have demonstrated the use of GNR in PTT for melanoma [26,27] and squamous cell carcinoma [25,28].

Recently, some groups have tried to combat some of the inherent difficulties of GNP use in vivo through supramolecular chemistry. Smaller nanoparticles are more suitable for cancer therapeutic applications as a result of their longer blood residence times and shorter biological half-lives. To this end, Cheng et al. have utilized photoactivable diazirine groups on gold nanosphere surfaces to allow spatiotemporal assembly of GNP aggregates suitable for photothermal therapy (Figure 1) [29]. Initially, the nanoparticles do not absorb NIR light. Following irradiation under 405 nm light, the nanospheres covalently cross-link through the diazirine groups, shifting their optical absorption spectra into the NIR region and allowing for selective spatiotemporal assembly of therapeutic nanoparticles in the target tissue.

Graphene oxide (GO) nanosheets have also been used in combination with PTT for cancer treatment. GO nanosheets have been demonstrated as effective photothermal absorbers [30]. Principally, Liu et al. have explored the use of GO in PTT [31]. GO-PEG nanosheets were also conjugated with the NIR fluorescent dye Cy7 for in vivo monitoring, showing that most of the nanosheets localized within the tumor and kidney. Following intravenous injection, local ablation of 4T1 murine breast cancer was achieved with low power densities (0.5 W/cm^2^) compared to similar concentrations of other nanoparticles. All mice survived following GO-PEG + laser treatment. Closely related to their other work, Liu et al. have also loaded GO nanosheets with Chlorin e6 to treat tumors using PTT-enhanced PDT [32]. Toxicologically, GO nanosheets exhibit dose-dependent cytotoxicity [33]. However, no significant abnormalities were observed by Liu et al. by histological examination following PTT with GO-PEG [31].

Closely related to GO nanosheets, carbon nanotubes, especially single-walled carbon nanotubes (SWCNT), have been utilized as photothermal absorbers for PTT. Both nanomaterials are based on a specific morphology of sp^2^ carbon graphene sheets. While GO nanosheets remain as one atom thick sheets that have a diameter of a few nanometers, SWCNTs are graphene sheets that have been rolled up and extend out as a tube. Because the nanotubes can be constructed with an ultrahigh length-to-diameter ratio, they are often considered to be one-dimensional nanowires. Like the production of GNR, SWCNT synthesis can be adjusted to tune the peak optical absorption of the material to a particular wavelength of light. When used appropriately, SWCNTs can serve as suitable photothermal absorbers for PTT.

Zhou et al. have demonstrated the use of SWCNT in PTT [34]. SWCNTs were conjugated to folate to target the tumor cells. Murine mammary EMT6 tumors were then injected with FA-SWCNT and irradiated with 980 nm NIR laser light at 1.0 W/cm^2^. It was seen that the damage was contained to the target tumor cells. Other groups have demonstrated similar effects using PEGylated SWCNT in epidermoid mouse cancer models [35] and a second murine mammary model, 4T1 [36]. Compared to GNR, biologically compatible SWCNTs could achieve effective tumor ablation using 10 times lower injected doses and lower laser powers [37].

Other nanoparticles have also been utilized as suitable photothermal absorbers in PTT. Copper sulfide (CuS) nanoparticles have been demonstrated as effective photothermal transducers in PTT [38], PTT-radiotherapy [39], and PTT-PDT [40]. CuS nanoparticles display minimal cytotoxic effects, similar to gold nanoparticles. Hessel et al. have shown the applicability of copper selenide nanocrystals for PTT [41]. The copper selenide nanocrystals were shown to have a higher photothermal transduction efficiency (22%) than gold nanorods (21%) and nanoshells (13%), indicating that they could be more suitable for PTT than comparable gold nanoparticles. Lastly, ultra-small black phosphorus quantum dots (BPQD) have also been used for this purpose [42]. The BPQDs were demonstrated to have a photothermal transduction efficiency of 24.8%, and the cytotoxicity of the material was reduced by PEG surface modification. As a result, BPQD could serve as an effective photothermal absorber for PTT.

Zhang et al. have developed Mo-based polyoxometalate (POM) nanoparticles which are capable of self-assembly under acidic and reductive tumor conditions [16]. It is a well-known effect that tumors often develop a preference for glycolysis that leads to an acidic microenvironment within the tumor as compared to standard physiological environments [43]. Zhang et al. have leveraged this effect by introducing a new nanomaterial paradigm capable of self-assembly under these conditions such that the materials would congest in the tumor and develop a stronger NIR absorbance for phototherapy. In addition, these materials develop a strong photoacoustic signal contrast for diagnostic purposes. The therapeutic effect of this material was demonstrated in vivo by ablation of 4T1 tumors.

## 3. Nanomaterials as Targeted Drug-Delivery Vehicles

Beyond acting as photothermal absorbers, nanomaterials can also be effectively utilized for precision-targeted drug delivery to enhance PTT. Nanomaterials have a diverse variety of surface chemistries and can be conjugated with molecular targeting agents like antibodies that allow them to target tumor tissue. While this is often used to enhance the specificity of thermal ablation, it can also be leveraged to synergistically deliver drugs directly to the tumor environment.

Zhang et al. have utilized graphene oxide nanosheets modified with PEG and the chemotherapeutic drug doxorubicin (DOX) to treat EMT6 mammary tumors in BALB/c mice [44]. Using 2W/cm^2^ continuous-wave laser irradiation at 808 nm, 80% of the mice in the complete treatment group had complete tumor ablation one day post-treatment with no tumor recurrence in the following 40 days. GO-PEG+laser and DOX alone had partial tumor ablation and slowed growth for the first week but failed to successfully destroy the entire primary tumor. Similarly, Ma et al. have constructed functionalized GO-PEG-DOX nanoparticles also loaded with iron oxide nanoparticles for simultaneous magnetic resonance imaging during combined photothermal-chemotherapeutic cancer treatment [45].

Zhou et al. utilized CuS nanoparticles as drug delivery vehicles in combined PTT-radiotherapy [39]. The nanoparticles were formed from solution containing non-radioactive and radioactive copper salts. When the nanoparticles are administered, the CuS nanostructure retains its photothermal traits while also emitting β—radiation to damage the local tumor tissue. The efficacy of the treatment was assessed in vivo on xenograft mice bearing Hth83 anaplastic thyroid carcinoma cancer model. The mice treated with PTT-radiotherapy had extended survival compared to the groups treated with nanoparticles alone, laser alone, radiotherapy alone, PTT alone, and the untreated controls.

Gold nanorods have also been utilized in PTT for their drug-delivery capabilities. Pandey et al. have synthesized carbon dot functionalized GNRs (C-dots@GNR) for photothermal applications [46]. Doxorubicin hydrochloride (DOX) was loaded onto the carbon dot surface via covalent and non-covalent pH-sensitive bonds. Following exposure to 808 nm laser light, the C-dots@GNR-DOX were effective at killing MCF 7 tumor cells in vitro. Another study performed by Guo et al. have developed hybrid chitosan nanosphere-GNR nanostructures loaded with the anticancer drug cisplatin [47].

Similarly, Li et al. have developed a GNR nanocomposite capable of delivering DOX to tumors for combined chemo-photothermal therapy (Figure 2) [48]. Nano-metallic oxide frameworks composed of zeolitic imidazolate framework-8 (ZIF-8) were synthesized around GNR to create a core-shell nanostructure. Various anticancer drugs could be loaded into the ZIF-8 shell, as demonstrated by DOX loading. Acidic pH and laser irradiation both lead to the decomposition of the core-shell nanostructure, releasing both the DOX into the local environment for selective chemotherapy and the GNR into the local environment for selective photothermal therapy.

Other forms of colloidal gold are also useful for cancer therapeutics. Elbialy et al. have developed magnetic gold nanoparticles (MGNPs) that can be targeted to the tumor region using an applied magnetic field [49]. In the work, they loaded DOX into the nanostructures for selective delivery of chemotherapy to the targeted region. By applying a 1 T neodymium magnet to the tumor surface for two hours, they were able to hold the nanoparticles in the tumor region. The study demonstrated that the targeting mode of the MGNP-DOX was capable of enhancing both the drug delivery to and photothermal effect in the tumor tissues.

Zhang et al. demonstrated the use of molybdenum disulfide (MoS_2_) nanosheets for drug loading and delivery to tumors via biological targets in conjunction with photothermal therapy [50]. In particular, MoS_2_ nanosheets were loaded with folic acid grafted to bovine serum albumin (FA-BSA) and DOX to target folic acid positive breast cancer. FA-receptor positive cell lines showed higher uptake of DOX, and the MoS_2_ nanosheets were demonstrated to release DOX and enhance the photothermal effect when irradiated by an NIR laser. Thus, the MoS_2_ nanosystems were capable of directly targeting FA-receptor positive breast cancer and delivering combined photo-chemotherapy.

## 4. Nanomaterials as Photoimmunological Agents

While drug-delivery is often applied to enhance the local tumor-killing effect of PTT, it can also be utilized to delivery immunological agents to the local tumor microenvironment to stimulate a systematic and long-term anticancer immune reaction. This paradigm shift enhances PTT to not only treat local tumors but also to treat advanced, metastatic tumors.

Based on current understanding, effective immunological treatment of cancer tumors requires three primary components: local tumor antigen release, immune cell recruitment, and immunological stimulation. PTT is effective in both destroying tumor cells to release antigens in the local microenvironment and instigating an inflammatory response to recruit immune cells into the region, and PTT is thus an apt tool for combination with immunotherapy [51,52]. As many of the tumor antigens are similar to those of the native healthy cells from the tissue which the cancer was derived, further immune cell stimulation is necessary before an effective treatment response can be achieved [53]. This is most often achieved using an immunoadjuvant drug (e.g., imiquimod). When all three components are added together, a systemic anticancer response can be stimulated, and metastatic cancers can be treated.

Chen et al. have pioneered a treatment called laser immunotherapy based on the use of photothermal laser irradiation and an immunoadjuvant (Figure 3) [18,54]. Small-molecule optical dyes such as indocyanine green (ICG) were initially used as a kind of nanoparticle to enhance the local photothermal effect. Zhou et al. have conjugated glycated chitosan (GC), an immunoadjuvant, to SWCNTs for nanomaterial-enhanced LIT [55]. SWCNT-GC has been demonstrated to colocalize within the mitochondria of the tumor cells, leading to enhanced photothermal tumor cell destruction due to mitochondrial disruption [56]. In vivo experiments have shown the efficacy of SWCNT-GC LIT on EMT6 metastatic mammary tumors [55]. All mice treated with laser + SWCNT-GC survived following treatment, compared to few survivors in the laser + SWCNT, laser + GC, and laser only groups. What was unique about the cured mice in the laser + SWCNT-GC group was that no mice formed tumors following rechallenge with EMT6 cells, whereas all other surviving mice formed tumors and died. This demonstrates that SWCNT-GC LIT can induce a systemic, long-term antitumor immune response to treat metastatic cancers.

Similar work has been done using alternative nanomaterials. Chen et al. have constructed poly(lactic-co-glycolic acid) (PLGA) nanoparticles loaded with imiquimod (IMQ) and ICG [15]. The results of the study demonstrated that the PLGA-IMQ-ICG nanoparticles could effectively transduce the laser light into heat, kill local tumor cells, and induce a systemic immune response in the metastatic mammary tumor model, 4T1. Their treatment was further enhanced through the use of a checkpoint inhibitor drug, anti-CTLA-4. Blocking CTLA-4 receptors reduce the role that T-regulatory cells play in inhibiting a systemic immune response. The study also demonstrated that administration of anti-CTLA-4 potentiated the photoimmunotherapy and formed an effective combination therapy.

Wang et al. have demonstrated the use of PEGylated SWCNTs with anti-CTLA-4 therapy to inhibit cancer metastasis (Figure 4) [57]. 4T1-bearing mice were injected with SWCNT-PEG and irradiated under 0.5 W/cm^2^ laser irradiation for 10 minutes. On days 1, 3, and 5 post-treatment, anti-CTLA-4 was injected via the tail vein. The survival of the mice was greatly enhanced by the combination SWCNT-PEG + anti-CLTA-4 treatment as compared to SWNCT-based PTT alone, surgical resection + anti-CTLA-4, and surgical resection alone. Additionally, by analyzing the lungs of the treated mice, it was found that the SWCNT-PEG + anti-CTLA-4 significantly reduced the number of lung metastases as compared to surgical removal of the primary tumors. This study suggests that potentially metastatic cancers traditionally treated by resection could see lower treatment failure rates when treated with photoimmunological therapy instead.

A similar treatment approach has been taken by Cano-Mejia et al. in treating a neuroblastoma model in vivo [58]. Prussian blue nanoparticles (PBNPs) were synthesized and injected into Neuro2a tumors. Tumors were then irradiated under 808 nm light at 1.875 W/cm^2^ for 10 min. On days 1, 4, and 7 after treatment, anti-CTLA-4 antibodies were systemically injected into the mice. The survival rates were comparable to other photoimmunological therapies: the PBNP + PTT + anti-CTLA-4 treatment group had 60% survival, the anti-CTLA-4 group had 10% survival, and there were no survivors in the untreated, PBNP + PTT alone, and PBNP alone controls. Of note, the authors also demonstrated that PBNPs exhibited pH-dependent stability, indicating that the nanoparticles should break down at physiological pH in the days following treatment.

Many other materials can be selected for use as either an immune adjuvant or as a photothermal absorber for use as photoimmunotherapeutic agents. DNA containing CpG sequences act as a TLR9 agonist, stimulating the maturation of APCs [59]. This has led Yata et al. to develop a GNR-DNA hydrogel for photoimmunotherapy of cancer cells [60]. As DNA can be loaded directly onto the GNR surface, hydrogel assembly can be mediated through the DNA interactions. Under near-infrared laser irradiation, the hydrogel structure began to break down, and the DNA containing CpG sequences are released. The efficacy of this material was demonstrated in vivo by treating EG7-OVA lymphoma tumor model-bearing mice.

## 5. Nanomaterials as Theranostic Tools

A critical facet to cancer therapy is accurate observation of the tumors themselves. This can prove challenging for some tumors as they can closely resemble the healthy tissue under many imaging modalities. Nanoparticles have been used to enhance the imaging process, either as a contrast agent in a traditional imaging modality or through specific signal generation (e.g., fluorescence). This can be a powerful tool in PTT, allowing for simultaneous therapeutics and diagnostics (termed ‘theranostics’) and reducing treatment complexity.

Both Ma et al. and Wang et al. have constructed iron oxide nanoparticle-modified graphene oxide nanoparticles (GO-IONP) for cancer theranostics (Figure 5) [45,61]. The GO sheets act as effective photothermal absorbers in the tumor tissue, and the iron oxide crystals act as contrast agents for MRI imaging. By depositing iron oxide nanocrystals onto graphene sheets followed by covalent bonding to branched PEG polymers, biologically-stable GO-IONPs were produced. Ma et al. demonstrated their therapeutic capabilities in vitro on 4T1 cells [45]. The GO-IONPs could act as effective photothermal absorbers, and their killing effect could be modulated by affecting the location of the nanoparticles through an external magnetic field. The diagnostic capabilities were determined in vivo on 4T1-bearing mice. MRI images of mice injected with GO-IONPs showed significantly decreased average MR signal generation in T_2_-weighted images than those without. Thus, the GO-IONPs could act as both a therapeutic agent and as a contrast agent for MRI imaging.

Wang et al. demonstrated the use of GO-IONPs in treating regional lymph node metastasis of pancreatic cancer [61]. A common site of metastasis that can lead to post-surgical complications in pancreatic cancer resection is the regional lymph nodes, the immediate repository of the draining lymphatics of the tumor tissue. Following injection, the GO-IONPs drain into the regional lymph nodes, and the lymph nodes can be mapped through MRI imaging. Once the locations of the relevant lymph nodes have been determined, a small incision was made into the skin of the mouse, and the lymph nodes were ablated using PTT. This treatment effectively demonstrated the considerable theranostic capabilities of GO-IONPs.

Liu et al. have developed a novel theranostic tool in bismuth sulfide nanorods (Bi_2_S_3_ NRs) that can be used for multimodal multispectral optoacoustic tomography (MSOT)/X-ray computed tomography (CT) imaging [62]. 4T1-bearing mice were injected with Bi_2_S_3_ NRs and irradiated with 808 nm laser light at 1 W/cm^2^, and the tumors were effectively ablated with no tumor remaining in any treated mouse after eight days. The sizable tumors were detectable by MSOT following intravenous injection of Bi_2_S_3_ NRs. Images from untreated mice show significantly less photoacoustic signal in the tumor region. Similarly, the tumor vasculature could be observed by CT post-injection.

Antaris et al. have demonstrated theranostic PTT with specific chirality SWCNT [63]. (6,5) Carbon nanotubes exhibit strong photoluminescence to NIR laser light. 4T1-bearing mice were injected with (6,5) Carbon nanotubes, and the SWCNTs remained within the tumor volume due to the enhanced permeability and retention effect of tumor tissues. Then, the photoluminescence was imaged by irradiating the tumors with 808 nm laser light at a power density of 0.14 W/cm^2^. The resulting emissions from 900 to 1400 nm were collected with an exposure of 100 ms. The tumors are clearly visible by visual inspection of the images. Following imaging, the tumors were also irradiated under 980 nm laser light at a power density of 0.6 W/cm^2^ for therapeutic purposes. Thus, specific chirality SWCNTs are intrinsically theranostic tools.

In addition, many organic conducting polymers have been introduced to enhance photothermal cancer therapy, such as polypyrrole (PPy) [64] and polyaniline [65]. Recent work on these materials has centered on enhancing the therapeutic efficacy of these materials. Jin et al. synthesized PPy nanoparticles based on polymerization around bovine serum albumin (BSA) polymers and conjugated ICG as a fluorescent probe and SP94, a hepatocellular carcinoma (HCC)-targeting peptide, to treat HCC in murine model [66]. The SP97 modified PPy-BSA-ICG showed highly specific uptake by tumor cells post-injection, and a strong fluorescence and photoacoustic signal for diagnostic purposes. Following NIR laser irradiation, complete local ablation of tumors was achieved only for the SP94 modified PPy-BSA-ICG, and the mice remained tumor free for the remaining three weeks of observation. When contrasted to the in vivo results of the PPy-BSA-ICG nanomaterial without SP94 where tumors were ablated and recurred within the observation period, this demonstrated the therapeutic efficacy of the targeting mechanism by SP94.

Theranostic applications of nanomaterials can also utilize photoacoustic (PA) imaging [16], positron emission computed tomography (PET/CT) imaging [67], and fluorescence imaging [68,69] for diagnostic purposes in combination with PTT. Many of the comparative advantages of the nanoimaging tools can be understood as the advantages of their respective imaging modalities. MRI contrast-enhancing nanomaterials offer strong diagnostic resolution at the cost of expensive procedure requiring specialized equipment. X-ray and PET/CT-based imaging offers deep tissue penetration but exposes the patient to damaging radiation in the process. Optical imaging modalities like fluorescence imaging do not utilize ionizing radiation; however, optical photons have low tissue penetration. Photoacoustic imaging ameliorates some of the issues with tissue penetration of optical imaging modalities, but it has less specific signal and is still unable to image deep tissues.

## 6. Conclusions

Nanomaterials have been demonstrated by many researchers to effectively enhance PTT-based cancer therapy through many mechanisms. With them, PTT has been expanded from a modality to ablate local tumors to a modality capable of treating local tumors and advanced metastatic cancers. In this work, we have reviewed the major nanomaterials used in PTT of cancer from a functional paradigm: nanomaterials as photothermal agents, drug-delivery vehicles, photoimmunological agents, and theranostic tools. Many of the first nanomaterials used for PTT such as GNRs and rGO have been seen as routine usage throughout the years; however, the field has advanced with many new exciting materials capable of advanced self-assembly, targeted drug release, and combination therapy approaches to treatment.

New insights and developments in nanomaterials science continue to push nanomaterial-enhanced PTT forward towards clinical application. However, much work must still be done before this can happen. New comparative studies between the various nanosystems can help to elucidate the optimal treatment regimen, and new biocompatibility studies can help to determine what qualifies as safe and appropriate usage of these materials. In addition, many materials have been developed that have expanded the functional tools available for cancer therapy, yet their biological properties, particularly regarding toxicity and fate after injection, are poorly understood. Until these materials have been proven to clear after treatment with no adverse effects, it is unlikely that they will see clinical application. With future studies, however, it is likely that the capabilities of nanomaterials to enhance cancer therapy will continue to improve. Many of the pre-clinical results of nanomaterial-enhanced PTT have demonstrated strong prospects to present new, low side-effect treatment options for patients in the future.

## Figures and Tables

**Figure 1 materials-12-00779-f001:**
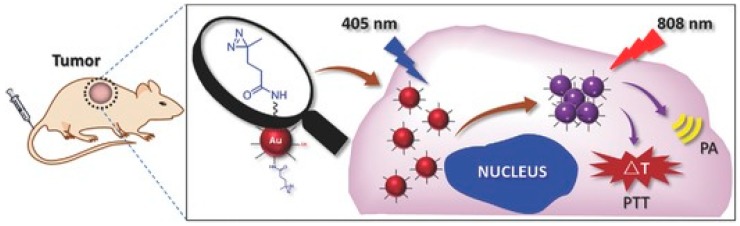
The use of supramolecular chemistry to engineer new nanomaterials in vivo. By assembling phototherapeutic nanoparticles in vivo as an aggregate of smaller nanoparticles, Cheng et al. are able to shorten the biological half-life of the nanomaterial and increase the biocompatibility. Reproduced with permission from [29], copyright © 2016, John Wiley and Sons.

**Figure 2 materials-12-00779-f002:**
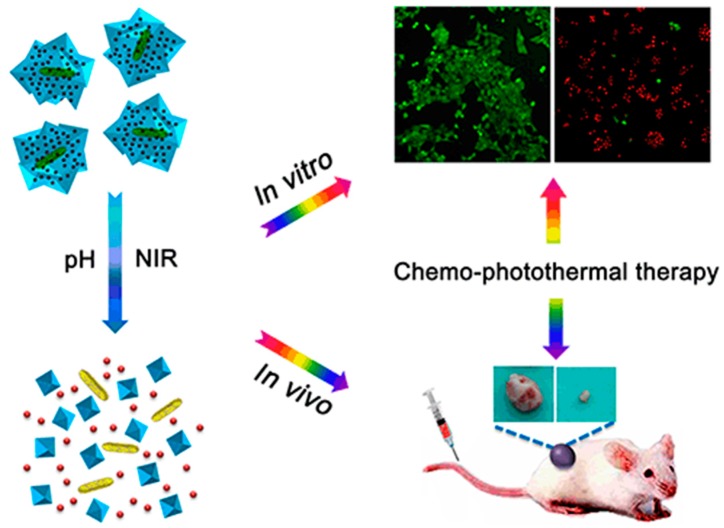
Crystalline zeolitic imidazolate framework-8 encapsulates doxorubicin hydrochloride and gold nanorods for synergistic chemo-photothermal therapy. The efficacy of chemo-photothermal therapy was demonstrated both in vitro and in vivo. Reproduced with permission from [48]. Copyright © 2017, Tsinghua University Press and Springer-Verlag GmbH Germany, part of Springer Nature.

**Figure 3 materials-12-00779-f003:**
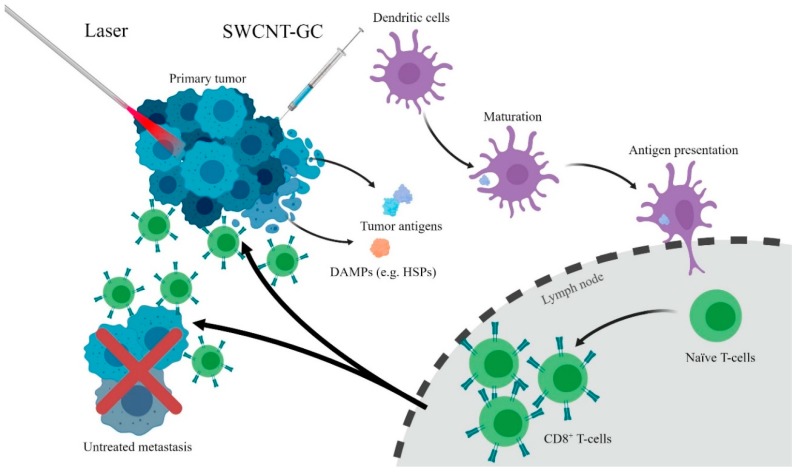
Depiction of laser immunotherapy. Tumors treated with Laser Immunotherapy release damage-associated molecular patterns (DAMPs), which, alongside glycated chitosan (GC), stimulate dendritic cell maturation and antigen presentation. This leads to activation and proliferation of CD8^+^ killer T-cells and a systemic antitumor immune response.

**Figure 4 materials-12-00779-f004:**
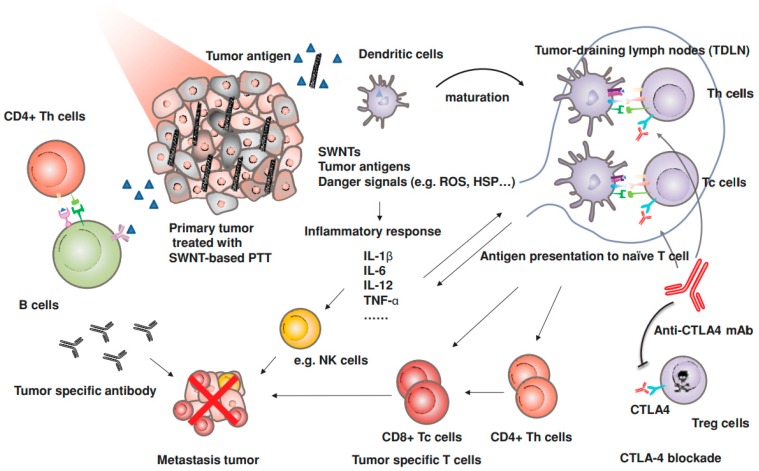
Proposed mechanism of nanomaterial-enhanced photoimmunological therapy. Tumors treated with SWCNT PTT can instigate a systemic immune response to treat metastatic cancers that can be potentiated through anti-CTLA-4 checkpoint inhibitor therapy. Reproduced with permission from [57]. Copyright © 2014 John Wiley and Sons.

**Figure 5 materials-12-00779-f005:**
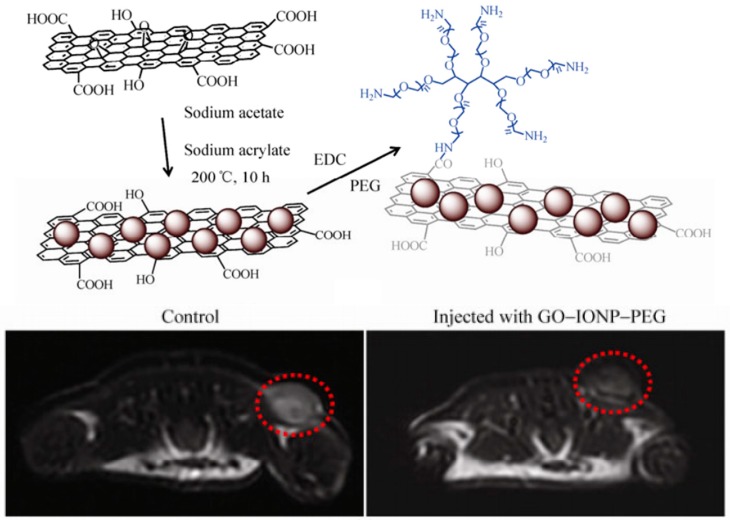
Iron oxide nanocrystal-modified graphene oxide nanoparticle (GO-IONP) synthesis and application as theranostic agent. The graphene oxide acted as an effective photothermal absorber while the iron oxide nanocrystals could simultaneously act as a contrast agent in MRI imaging. Reproduced with permission from [45]; Copyright © 2012 Springer Nature.

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
