# Peer review of "Nanomaterial Applications in Photothermal Therapy for Cancer"

_materials, 2019, doi:10.3390/ma12050779_

Round 1
Reviewer 1 Report
Chen and co-workers in the manuscript titled “Nanomaterial applications in photothermal therapy for cancer” summarized the recent development of nanoparticulate agents for photothermal therapy. Nanomaterials such as gold NPs, GO nanosheets, SWCNTs, etc. have been well concluded. The PTT combined with another therapeutic method such as chemotherapy and immunological treatment was introduced together with a section of the nanoparticulate theranostics. Overall, the review is comprehensive. Some concerns should be addressed before publication.
1. Many other reported photothermal nano agents should also be introduced (e.g., semiconducting polymer NPs, Polyoxometalate clusters, etc.)
2. For the theranostic part, can the authors please enrich this section since many other imaging methods has been used combing the PTT (e.g., PA, PET, upconversion luminescent imaging, etc.)
Reviewer 2 Report
This could be a mini-review.
Although, author explained the NMs as drug-delivery vehicles, the key part of targeting of malignant cells using NMs is missing as according to the title. Please include the missing part.
Inclusion of more strong discussions over toxicity could support the importance of NMs in PTT.
Importantly, details about the fate of NMs after PTT would be a great impact point of this article.
Provide a table of advantages and disadvantages of PTT therapy of each NMs.
Include detailed summary and outlook/future perspectives of this review article.
Minor,
Typo errors (ex.carful)
Reviewer 3 Report
In the presented work, the authors have studied applications of nanomaterials to enhance cancer therapies specifically as photothermal absorbers, drug delivery vehicles, photoimmunological agents, and theranostic tools. It worth to noted that it is a very interesting research area with a large impact on today’s quality of our life. However, the paper should be corrected in order to improve the paper's quality.
Authors should be inserted a little bit details about described processes e.g. range of value of temperatures of photothermal therapy of presented absorbers - it is known that tumor damage by laser therapy typically commences at 41°C.
Authors pointed out that the paper presents a review of current applications of nanomaterials (e.g. line 18-19). Unfortunately in the work is a small value of references to some newest scientific papers published within five years (taking to account submission date) related to the taken topic. Are authors sure that newer publications don't exist?
Round 2
Reviewer 2 Report
The MS amended the requested revision with major changes which could be useful for readers in this research area.
This MS is to be ACCEPTED in its present format.